# Romantic Attraction and Substance Use in 15-Year-Old Adolescents from Eight European Countries

**DOI:** 10.3390/ijerph16173063

**Published:** 2019-08-23

**Authors:** András Költő, Alina Cosma, Honor Young, Nathalie Moreau, Daryna Pavlova, Riki Tesler, Einar B. Thorsteinsson, Alessio Vieno, Elizabeth M. Saewyc, Saoirse Nic Gabhainn

**Affiliations:** 1Health Promotion Research Centre, School of Health Sciences, National University of Ireland Galway, University Road, H91 TK33 Galway, Ireland; 2Institute of Psychology, ELTE Eötvös Loránd University, Izabella utca 46, 1064 Budapest, Hungary; 3Department of Interdisciplinary Social Science, Faculty of Social and Behavioural Sciences, Utrecht University, P.O. Box 80.140, 3508 TC Utrecht, The Netherlands; 4Centre for the Development and Evaluation of Complex Interventions for Public Health Improvement, Cardiff School of Social Sciences, Cardiff University, 1–3 Museum Place, CF10 3BD Cardiff, UK; 5Independent Researcher, 1180 Brussels, Belgium; 6Department for Monitoring and Evaluation of Social Projects, Ukrainian Institute for Social Research after Oleksandr Yaremenko, 26 Panasa Myrnogo Str., Of. 211, 01011 Kyiv, Ukraine; 7The Department of Health Systems Management, Ariel University, Ramat HaGolan St 65, 40700 Ariel, Israel; 8School of Psychology, Faculty of Medicine and Health, University of New England, Armidale, NSW 2351, Australia; 9Department of Developmental and Social Psychology, University of Padova, 8 via Venezia, 35131 Padova, Italy; 10Stigma and Resilience Among Vulnerable Youth Centre (SARAVYC), School of Nursing, University of British Columbia, T222-2211 Wesbrook Mall, Vancouver, BC V6T 2B5, Canada

**Keywords:** adolescents, romantic attraction, same-gender attraction, both-gender attraction, sexual minority youth, substance use, alcohol consumption, drunkenness, tobacco, cannabis, HBSC

## Abstract

Sexual minority youth are at higher risk of substance use than heterosexual youth. However, most evidence in this area is from North America, and it is unclear whether the findings can be generalized to other cultures and countries. In this investigation, we used data from the 2014 Health Behaviour in School-aged Children (HBSC) study to compare substance use in same- and both-gender attracted 15-year-old adolescents from eight European countries (*n* = 14,545) to that of their peers who reported opposite-gender attraction or have not been romantically attracted to anyone. Both-gender attracted, and to a lesser extent, same-gender attracted adolescents were significantly more likely to smoke cigarettes, consume alcohol, get drunk and use cannabis, or be involved in multiple substance use in the last 30 days compared to their opposite-gender attracted peers. Those adolescents who have not been in love had significantly lower odds for substance use than all other youth. The pattern of results remained the same after adjusting for country, gender and family affluence. These findings are compatible with the minority stress and romantic stress theories. They suggest that sexual minority stigma (and love on its own) may contribute to higher substance use among adolescents in European countries.

## 1. Introduction

Many young people who identify as Lesbian, Gay, Bisexual (LGB), other sexual or gender minority (for example Queer, Transgender or Intersex), or report being attracted to same- or both-gender partners, have poorer health than their peers who identify as heterosexual, cisgender or as exclusively attracted to members of the opposite gender [1,2]. The studies show a large variation in the use of (biological) sex or (socially constructed) gender. They also employ various sexual identity terms or classify youth based on other dimensions of sexual orientation, such as gender of sexual or love partner(s). In this study, we use the term ‘gender’ to describe whether the respondents identified themselves as boys or girls. The ‘sexual minority youth’ (SMY) term is used, as this is the most inclusive, unless we refer to studies that used more specific terminology (such as LGB). 

Extensive research indicates that SMY are more likely to engage in substance use [3,4]. However, the validity of the evidence is limited by the fact that most investigations have been conducted in North America. There are just a few sporadic observations from other countries, and cross-cultural comparisons are largely missing. This study aimed to describe and compare substance use frequency across patterns of romantic attraction, in nationally representative samples of 15-year-old adolescents from eight European countries and regions with various geographical location, history, and levels of tolerance towards sexual minorities. 

### 1.1. Tobacco Use 

While LGB youths appear to start smoking at a later age than the general population, compared to their heterosexual peers they are significantly more likely to use various (and multiple) tobacco products, as well as to report smoking in the past month, or being current smokers [5,6,7,8]. Smoking patterns were influenced by sexual identity, gender, race or ethnicity, and their interactions [8,9,10]. In some studies, significant differences were found between sexual minority girls and boys, or bisexual youths and those identifying as lesbian or gay. Such findings indicate the importance of mapping the relative risk of SMY boys and girls separately.

Cross-sectional and longitudinal studies have concluded that psychological distress in SMY was associated with smoking [11,12]. Sexual minority adolescents were significantly more likely to report smoking in the past year compared to heterosexual youth in a large-sample U.S. national prospective cohort study, after adjusting for gender, age, race/ethnicity, and family income. However, SMY youth living in states where the social environment was less stigmatizing toward LGB people had a significantly lower relative risk for smoking than those who lived in a state imposing stronger structural stigma. The stigmatizing environment did not have a differential effect on heterosexual youth [13]. This indicates that besides micro-environmental influences, macro-level societal indicators may also be associated with substance use in SMY. Therefore, it is important to investigate cross-cultural variations in the associations between sexual minority status and substance use or other risk behaviors, ideally including sexual minority young people from various countries and cultures.

### 1.2. Alcohol Consumption and Drunkenness

Sexual minority adolescents are more likely than heterosexual youth to drink alcohol and get drunk [3,14]. LGB young people report earlier alcohol initiation and sharper drinking trajectories into adulthood than heterosexual youth [15]. The experience of sexual minority belonging in adolescence may shape alcohol-related behaviors in later age [16]. Some argue that consuming any amount of alcohol and excessive drinking (i.e., heavy episodic drinking) may diverge, for instance for cultural reasons [17], therefore they should be examined separately.

While there was a general decline in adolescent alcohol use in the United States and Europe over the last decade [18], the alcohol-related disparities between heterosexual and SMY have remained stable or even widened [19]. More recent findings demonstrate that pluri-sexual males (having both-gender sexual partners or identifying as bisexual) have higher risk for earlier onset and persistent use of alcohol than those with monosexual behavior (having exclusively opposite- or same-gender partners) or identifying as heterosexual or gay [20].

### 1.3. Cannabis Use

SMY are at an increased risk of cannabis use [21,22,23]. A systematic review of Canadian studies revealed that cannabis use is consistently higher in SMY than in heterosexual respondents, with sexual minority young men and bisexual youth having greater risk than young women or those identifying as lesbian or gay [24]. Furthermore, trends observed in cannabis use in SMY are similar to those in alcohol and tobacco use. Between 1999 and 2013, a population-based US study found that while cannabis use had decreased overall, the disparities between some SMY subgroups (in lesbian and bisexual females) and their heterosexual counterparts remained stable [25].

Potential explanatory pathways for the association between SMY status and cannabis use are comparable to those of alcohol and tobacco use. Internalized homophobia and community connectedness were both positively associated with cannabis use in LGB young people [26]. These effects may be attributed to minority stress, and greater community connectedness may be associated with greater conformity to social norms within the LGB community that are more permissive toward substance use.

### 1.4. Minority and Romantic Stress: Explanation for Different Types and Combinations of Substance Use?

In addition to the single substance studies cited above, there is a large corpus of evidence on multiple or poly-substance use and the association with mental health outcomes in SMY. These studies examine alcohol, tobacco and cannabis or other drugs [20,25,27], alcohol and cigarettes [28], drugs and alcohol [29,30], tobacco, methamphetamine use and suicidal ideation [31], or meta-analyses where different types of substance use were pooled [3,4].

In a systematic review of 18 studies [3], it was found that LGB youth were around three times more likely, compared to their heterosexual peers, to be involved in any type of substance use. The effects were larger in bisexual compared to lesbian/gay young people, and in females compared to males. When one large-effect size study was removed from the pool, no significant differences were observed between studies conducted in the United States or elsewhere, which suggests that the disparity may be universal across different countries and cultures. Another systematic review of 12 studies revealed that the strongest risk factors for substance use (smoking cigarettes, consuming alcohol, cannabis, cocaine and ecstasy) in SMY were LGB-related or general victimization, lack of supportive environments, psychological stress, internalizing/externalizing behaviors, negative responses to coming out, and housing status [4].

The disparities between SMY and heterosexual youth’s substance use can be explained by the minority stress theory which argues that experiences of discrimination, victimization, and stigma are prevalent due to a pervasive homophobic culture [32,33]. The existing literature points out that sexual orientation-based bullying and harassment at school contributes to SMY disparities in all forms of substance use. Hatzenbuehler’s [34] extension of minority stress suggests that due to stigmatization, sexual (and gender) minority individuals experience chronic stress, which in the long term may lead to deficits in emotion regulation and negative affect. To cope with these, sexual minority individuals may turn to alcohol (and other substance) use [4,12,35].

Consistent with the ecological framework provided by minority stress theory, it is important to examine factors that may predict substance use, particularly societal attitudes and policies regarding sexual minority communities and individuals. Although attitudes toward sexual minorities also are changing in several parts of the world, there are still many countries with strong anti-LGB policies or cultural norms. The negative effects of stigma and discrimination on sexual minority individuals’ health, including minority stress, depression, and fear of seeking help are well-documented. However, most of the evidence is from North America [36]. The question remains whether these findings can be generalized to other countries and cultures (i.e., in the European region), given the large variation in societal attitudes, tolerance and acceptance towards gender and sexual minority individuals and issues within Europe [37,38].

The countries involved in our study represent large variation both geographically (from Iceland to North Macedonia), historically (from traditionally Capitalist countries such as Belgium, England, France and Switzerland to post-Communist countries as Bulgaria and Hungary), and in terms of tolerance towards sexual minorities. The latter can be demonstrated by the International Lesbian Gay, Bisexual, Trans and Intersex Association’s (https://www.ilga-europe.org/rainboweurope) Rainbow Score, a composite measure reflecting the legal situation and acceptance of gender and sexual minorities in different countries, ranging from 0 (gross violations of human rights) to 100 (full respect of human rights, full equality between sexual and gender minority and heterosexual and cisgender individuals). In the eight countries or regions involved in the present study, the Rainbow Score in 2014, when the data were collected, ranged from 13% in North Macedonia to 82% in the United Kingdom [39].

In a nine-country investigation of substance use in 16–35-year-old LGB and heterosexual individuals, Demant and colleagues [40] argue that cross-cultural comparisons in this area are important because cultural norms and attitudes towards both substances and sexual minority identities show considerable variations across countries with liberal versus more conservative policies and regulations. Until such studies are replicated, we cannot conclude that higher frequency of substance use in SMY is a universal phenomenon. Therefore, in this study we aimed to explore the associations between SMY and different substance use behaviors across different European countries and regions.

Another potential explanation, partly overlapping with the minority stress model, is that love, irrespective of the gender of the partner(s) with whom a young person is in love with, may be associated with stress on its own. Indeed, a cross-cultural study conducted in 17 countries found that adolescents experienced stress related the romantic relationships, especially in Mid- and South-European countries. Overall, around 20% of the adolescents used externalizing coping strategies, such as alcohol and drug use, to cope with these stressors [41]. This prompts the notion that maybe not just same-or both-gender attracted adolescents may be at elevated risk of substance use, but *anyone* who are in love may be at higher risk than those who are not being in love.

### 1.5. Dimensions of Sexual Orientation

The number of young people with same-gender attractions far exceeds those who engage in same-gender sexual behavior or who identify as lesbian, gay or bisexual. This is consistent with findings from large-scale nationally representative studies with adults, where the proportion of individuals with same- or both-gender attraction was much larger than those who identified as LGB [42]. A population-based study in the United Kingdom demonstrated substantial diversity between identity, behavior and attraction in sexual minority adults [43]. Studies have also varied on how they categorized SMY (identity, behavior or attraction), and whether they separated mono- and plurisexual youth. In one study, respondents as young as 9–10 years old were asked whether they consider themselves to be lesbian, gay or bisexual [44]. While acknowledging the importance of all dimensions of sexual orientation, we argue that asking whether adolescents are attracted to girls, boys or both-gender partners may be easier for young people to answer, be more accurate, and can be used to subsequently categorize SMY based on same- or both-gender romantic attraction. This approach may be developmentally more appropriate than employing the identity labels of sexual orientation [45]. Relying on sexual identity as a classifier for SMY may ‘mask’ or eliminate those young people who are still exploring their sexuality, have same- or both-gender attraction, but do not identify as LGB.

Health disparities in SMY can be found when respondents are classified by same- or both-gender romantic attraction. In a nationally representative study of U.S. adolescents [46], boys romantically attracted to both-gender partners smoked more cigarettes, were more likely to have consumed alcohol while being alone, to have been drunk, and to use illegal drugs (including cannabis) compared to those who had been attracted to the opposite gender. Girls attracted to their same- or both-gender peers were more likely to smoke cigarettes, have been drunk, and have used cannabis or other drugs compared to opposite-gender attracted females. However, their conclusion was that SMY or certain subgroups within this category had a greater risk of substance use than heterosexually identifying or exclusively opposite-gender attracted youth. Therefore, in the current study we anticipate finding significant gender differences in the associations being investigated.

Another U.S. adolescent study demonstrated that sexual identity (i.e., defining oneself as LGB) and sexual behavior (i.e., having exclusively same- or both-gender partners) explained unique and significant sources of variability in tobacco and methamphetamine use and suicidal ideation [31]. In another investigation, same- and both-sex romantic attraction and romantic relationship status were associated with various risk behaviors such as the number of cigarettes smoked in the past month, being drunk in the past year, and cannabis or other drug use [46]. When adolescents were categorized into SMY based not on their identity but either on a history of same-gender attraction or sexual behavior, a sharper increase was observed in their cigarette and cannabis use than in those adolescents with heterosexual identity (or opposite-gender attraction or behavior) [47]. These findings demonstrate that apart from identity, other dimensions of sexual orientation (i.e., behavior or romantic attraction) may also be associated with higher incidence of risk behaviors.

Based on these considerations and empirical evidence, in the present study, romantic attraction will be used to classify sexual minority adolescents and separate adolescents reporting being in love with any gender partners from those who have not been in love.

### 1.6. Aims of the Present Study

We aimed to describe and compare substance use frequency across patterns of romantic attraction, in nationally representative samples of 15-year-old adolescents from eight European countries and regions.

Romantic attraction was operationalized by an item on whether the respondent had already been in love, and if yes, whether the partner who they felt love for was a girl(s), boy(s), or both- a boy and a girl [45]. This approach is in line with the notion that romantic attraction and love are conditional to each other [48]. The responses, combined with the gender of the respondent, enabled us to categorize opposite-, same-, or both-gender attracted respondents, those who have not been attracted to anyone, or who have not responded to the love item. Contrary to most studies that concentrate on those with any type of attraction or sexual identity, we also measured the prevalence of substance use in those who reported not having been in love or who did not respond to this item. Based on previous findings from the literature, we hypothesized that same- and both-gender attracted young people will have significantly higher odds of cigarette smoking, drinking alcohol, being drunk, and cannabis use than their opposite-gender attracted peers or those who reported not having been in love. Our other hypothesis is that those young people who report being in love (with any gender partners) will have higher odds of substance use than those who have not been in love. We anticipated that despite cultural differences in the prevalence of these risk behaviors, higher incidence of substance use will be found in same- and both-gender attracted young people (and those who report being in love) across different countries and regions. Given the differences between sexual minority boys and girls found in many studies, analyses were stratified for gender.

An additional aim was to assess involvement of SMY in multiple risk behaviors (any two or all three of cigarette smoking, alcohol consumption, or cannabis use in the last 30 days). We hypothesized that youth reporting attraction to same- or both-gender partners will be more likely to be involved in using more than one type of substances than those who are exclusively attracted to opposite-gender partners or reported not having been in love.

## 2. Materials and Methods

Data was collected within the 2014 survey round of the Health Behaviour in School-aged Children (HBSC) study, a World Health Organization collaborative cross-national epidemiological study. The HBSC study investigates health-related behaviors and related psychosocial contextual factors in nationally representative samples of 11-, 13- and 15-year-old school children, in four-year study cycles across more than forty countries, covering the geographical areas of Europe, North America, and former Soviet Republics. There were 42 countries that collected data as part of the HBSC international survey in 2014. Out of these, data from eight countries and regions are featured in this paper. A detailed description of HBSC methodology is provided by Inchley et al. [49] and Currie et al. [50]. In HBSC, a survey questionnaire is employed containing (1) items administered in each participating countries in the same format (‘mandatory’ items), (2) items following the same format, but the national team decides if they will be administered in the questionnaire (‘optional’ items) and (3) items that are relevant for the health of young people in the given country (‘national’ items). In the present study, substance use was monitored using mandatory items, while romantic attraction was measured by an optional item. As such, the measure of romantic attraction was included in the national surveys if the research team in the given country or region considered investigating the health of sexual minority youth substantially important. The methodology used by HBSC is described at http://www.hbsc.org/methods/index.html, and details on data access are provided at https://www.uib.no/en/hbscdata.

### 2.1. Sample

Schoolchildren in the 15-year-old age group from eight countries and regions (French Belgium, Bulgaria, Switzerland, England, France, Hungary, Iceland, and North Macedonia) where the national HBSC Research Team included the measure on romantic attraction (see below, Section 2.2) in their national survey. The raw sample contained data from 14,545 respondents (mean age: 15.55 years, *SD* = 0.33, range: 14.58–16.50, percentage girls: 49.8). Listwise deletion was employed (for all predictor, outcome and sociodemographic control variables) to determine the number of respondents featured in the final statistical models. There were 13,504 respondents (92.8%) in the cigarette smoking model; 13,440 respondents (92.4%) in the alcohol consumption model; 13,471 respondents (92.6%) in the drunkenness model; 12,109 respondents (83.3%) in the cannabis use model; and 13,580 respondents (93.4%) in the multiple substance use model. The characteristics of the sample are displayed in Table 1. Since the most respondents were featured in the multiple substance use model, the distribution of the Love item and sociodemographic variables are given for this headcount (*n* = 13,580).

### 2.2. Ethical Considerations

In each country, the HBSC research team sought ethical approval from local or national higher education or health authorities: Boards of School Networks of the Brussel-Wallonia Federation (French Belgium), Ministry of Education and Science (Bulgaria), University of Lausanne, Cantonal Commission for Ethics for the Research on Human Beings (Switzerland), University of Hertfordshire, Ethics Committee for Studies Involving Human Participants (England), Ministry of Education and the French National Commission of Computer Science and Freedom (France), Scientific and Research Ethics Committee of the Medical Research Council (Hungary), Icelandic Data Committee (Iceland), and the Ministry for Education and Ministry for Health (North Macedonia). In the eight countries and regions involved in this study, pupils (as well as their parents and the schools) gave informed consent to participate in the study. Before administering the questionnaire, respondents were instructed that responding to any question or the whole questionnaire was entirely voluntary, and they could withdraw at any time. The questionnaires were anonymous and treated as confidential. Our research procedures are following the WHO Standards and operational guidance for ethics review of health-related research with human participants (https://www.who.int/ethics/research/en/).

### 2.3. Measures

*Romantic attraction* was measured by a standardized item “Have you ever been in love with…”, response options being “A girl or girls”, “A boy or boys”, “Both girls and boys”, “I have never been in love”. Girls who reported being in love with boys, and boys who reported being in love with girls were categorized into the opposite-gender love group, while girls who reported being in love with girls and boys reporting love for boys were categorized into the same-gender love group. Respondents reporting being in love with both girls and boys were categorized into the both-gender love group. A fourth group consisted of those respondents reporting having never been in love, while the fifth category included those who did not answer the item. The development and basic descriptive statistics for the question are reported elsewhere [45].

*Substance use:* Four standardized items were used to measure the frequency of substance use in the last 30 days [50,51]. “On how may days (if any) have you smoked cigarettes (tobacco) in the last 30 days?”, “On how many days (if any) have you drunk alcohol in the last 30 days?”, “Have you ever taken cannabis (hashish, grass, pot) in the last 30 days?” with response options being “Never”, “1–2 days”, “3–5 days”, “6–9 days”, “10–19 days”, “20–29 days”, “30 days (or more)”. “Have you ever had so much alcohol that you were really drunk in the last 30 days?”, with response options being “Never”, “Yes, once”, “Yes, 2–3 times”, “Yes, 4–10 times”, “Yes, more than 10 times”. In line with methodological recommendations and reporting practice of the European School Survey Project on Alcohol and Other Drugs (ESPAD) for fifteen-year-olds [51], the four substance use variables were dichotomized into reporting never having used the given substance (never being drunk) *versus* ever. We created a dichotomous variable to express multiple substance use. If the respondent reported any two of cigarette smoking, alcohol consumption and cannabis use in the last 30 days, they were categorized into ever being involved in multiple substance use.

*Gender and age:* Respondents were asked to indicate whether they are a boy or a girl, as well as to report their date of birth (month/year).

*Socioeconomic status* was measured by the Family Affluence Scale (FAS), a six-item composite measure developed by the HBSC network [52,53,54]. FAS measures material family wealth as an indicator of socio-economic position. It asks about concrete possessions (i.e., number of family cars; computers), characteristics of the home (i.e., having a bedroom for one own; number of bathrooms; owning a dishwasher), and the number of family holidays in the last year. The scores are summed up. The absolute Family Affluence Scale scores (0 = lowest affluence, 13 = highest affluence) were then transformed into a ridit-based trichotomous variable separating children from families within the lowest 20%, the medium 60%, and the highest 20% affluence categories [49].

### 2.4. Statistical Analysis

Data analysis was carried out in IBM SPSS Statistics for Windows, version 25.0 (IBM Corp., Armonk, NY, USA). First, descriptive analyses were conducted for the overall sample and broken down into categories of romantic attraction. Chi-square tests were computed along with Cramér’s *V* effect sizes to check for potential associations between romantic attraction and the sociodemographic and substance use variables. Uni- and multivariate binary logistic regression models were built to map the odds of substance use in adolescents belonging to other romantic attraction categories, compared to those who reported (exclusively) opposite-gender love.

Univariate models were constructed to obtain crude odds ratios (COR) of substance use in different romantic attraction groups. The reference was the group reporting opposite-gender love. Then country, gender and relative FAS grouping were added to the models to obtain adjusted odds ratios (AOR). French Belgium, boys, and adolescents belonging to the lowest family affluence group were set as reference categories. Multivariate analyses were carried out for the entire sample and stratified for gender. Wald statistics indicated that each predictor variable made a significant contribution to the models (*p* ≤ 0.04). Model fit was examined. In many cases, the Chi-square tests indicated poor fit (*p* > 0.05), which may be a result of the large (overall) sample size and the imbalance between the compared subgroup sizes. This does not necessarily mean that the model should be discredited [55,56]. No collinearity was observed in the predictor variables. To test the potential confounding effect by interactions between the predictors, we constructed models that included two-way interactions, but these did not improve model fit.

It was at the discretion of the national HBSC teams to decide whether they would weight their data to correct imbalances in the composition of the sample. Data were not weighted if the characteristics of the actual sample corresponded to those of the national sampling frame (e.g., gender or family affluence distribution). The only exception to this was France. It means that from the eight national data sets included in this analysis, weighting was only applied to the French data. Therefore, we have used a weight variable with actual values for the French data and set to 1 for data from other countries.

The HBSC study uses classrooms as sampling units. To check whether cluster-based sampling method impacted the results, we have carried out the analyses using the Complex Samples function in SPSS. Design effects in the multivariate models, indicating the extent to which clustering effect needs to be corrected, were not substantially different from 1 (0.98 ≤ DEFF ≤ 1.24), indicating that clustering had a negligible impact. Therefore, the analyses have not been adjusted for cluster sampling.

## 3. Results

The number of respondents in each substance use group in the binary logistic models was determined by how many answered the given substance use item. As Table 2 shows, most respondents reported on the frequency of smoking item in the last 30 days (*n* = 13,504), while a lower number answered the items on alcohol consumption (*n* = 13,440), drunkenness (*n* = 13,471), and cannabis use (*n* = 12,109). For the multiple substance use model, all responses featured in any two of the single substance use models were collapsed (*n* = 13,580).

### 3.1. Love and Sociodemographic Characteristics

There was a significant association between love and country: *χ*^2^(28) = 1625.87, *p* < 0.001, but with a low effect size: *V* = 0.173 (Table 1). Love was associated with gender of the respondents: *χ*^2^(4) = 87.30, *p* < 0.001, but with a low effect size: *V* = 0.080. More girls reported same-gender love than boys, and the difference was even larger in the case of both-gender love. Girls were also more likely than boys to report not having been in love, but they were less likely than boys not to respond to the item. Love was also associated with family affluence: *χ*^2^(8) = 36.46, *p* < 0.001, but the effect size was negligible: *V* = 0.037.

### 3.2. Romantic Attraction and Substance Use

The prevalence of substance use across different attraction patterns is displayed in Table 2. Respondents reporting both-gender love reported the highest prevalence for each substance use: Cigarette smoking (33.6%), drinking alcohol (51.2%), being drunk (25.1%), and cannabis use in the last 30 days (20.6%). They also reported the highest rate of engagement in multiple substance use in the last 30 days (30.2%). Across all these romantic attraction groups, the lowest rates of engagement with substance use (apart from alcohol use) was reported by those who have never been in love.

### 3.3. Cigarette Smoking across Romantic Attraction

The univariate models indicated that same-gender and both-gender attracted respondents were significantly more likely to smoke in the last 30 days, while those not having been in love were less likely to report this behavior compared to opposite-gender attracted respondents (Table 3). Adjusting the model for country/region, family affluence, and gender did not substantially change the odds (the full model is displayed in Appendix A). Compared to opposite-gender attracted adolescents, those who had been in love with both-gender partners had odds of 2.3, while same-gender attracted had odds of 1.9 for smoking. Those not having been in love had significantly lower odds (AOR = 0.4). Those who did not respond to the love item had statistically similar odds of smoking to those who were opposite-gender attracted. Gender-stratified analyses demonstrated that both-gender attracted boys and girls had somewhat higher odds of smoking (AOR = 2.8 and 2.1, respectively) than the same-gender attracted boys and girls (AOR = 2.4 and 1.6, respectively), but these differences were not statistically different. Those who reported never having been in love were significantly less likely to smoke (boys’ AOR = 0.5; girls’ AOR = 0.4) compared to opposite-gender attracted youth.

### 3.4. Alcohol Consumption across Romantic Attraction

Same-gender attracted youth did not have higher odds of alcohol consumption in the last 30 days than those reporting love with opposite-gender partners, but both-gender attracted youth had significantly higher odds, whereas those not attracted and non-responders had significantly lower odds (Table 3). The unadjusted model and model adjusted for country/region, family affluence, and gender yielded a similar pattern (the full model can be found in Appendix A). Compared to their opposite-gender attracted peers, those who reported love for both-gender partners had odds of 1.8 for alcohol consumption, while those reporting never having been in love or not responding to the love item had odds of 0.5. However, analyses stratified for gender showed that only same-gender attracted boys (AOR = 1.7) and both-gender attracted girls (AOR = 2.2) were significantly more likely to have had alcohol in the last 30 days. Both boys and girls who reported never being in love or who did not respond to the item on love had significantly lower odds of alcohol consumption.

### 3.5. Drunkenness across Romantic Attraction

In the univariate model, same- and both-gender attracted youth were significantly more likely to report drunkenness in the last 30 days than opposite-gender attracted respondents, while those who had been in love less likely (Table 3). The same pattern was found in the multivariate model adjusted for country/region, family affluence, and gender (the full model can be found in Appendix A). Compared to opposite-gender attracted youth, those reporting same-gender attraction had of 1.8 times, and both-gender love odds of 2.2 (both significantly higher) for drunkenness, while never attracted (AOR = 0.5) and non-responding youth (AOR = 0.7) had significantly lower odds. Same-gender attracted boys were significantly more likely to report drunkenness (AOR = 2.4) but same-gender attracted girls were not (AOR = 1.4). No gender differences were observed in the other groups. 

### 3.6. Cannabis Use across Romantic Attraction

In the univariate model, same- and both-gender attracted adolescents were significantly more likely, while those who had not been in love were significantly less likely to report cannabis use in the last 30 days compared to their opposite-gender attracted peers (Table 3). After adjusting for country/region, family affluence, and gender, a similar pattern was observed (the full model can be found in Appendix A). Compared to those reporting opposite-gender love, those who had been in love with same-gender partners had odds of 2.2 and those in love with both-gender partners had odds of 3.6 (both significantly higher) of reporting cannabis use, while those who had never been in love had significantly lower odds (AOR = 0.6); the odds of non-responders were not significantly different from their opposite-gender attracted peers. Same-gender attracted boys’ odds were significantly higher (AOR = 2.9) but same-gender attracted girls had similar odds (AOR = 1.6) for cannabis use as their opposite-gender attracted peers. Both-gender attracted boys and girls had higher odds than those reporting opposite-gender attraction (boys’ AOR = 4.1, girls’ AOR = 3.2), whereas never having been in love was associated with significantly lower odds in both boys (AOR = 0.7) and girls (AOR = 0.5).

### 3.7. Multiple Substance Use across Romantic Attraction

The comparison of participants reporting multiple substance use among those with different patterns of attraction yielded analogous results to those found with use of single substances (Table 4). In both the univariate model and the model adjusted for country/region, family affluence and gender (the full model can be found in Appendix A), same- and both-gender attracted adolescents had higher odds of using any two or all three substances than opposite-gender attracted young people, while not being in love was associated with significantly lower odds. Non-respondents had a similar likelihood to their opposite-gender attracted peers of reporting multiple substance use. However, in the gender-stratified analyses, only same-gender attracted boys had significantly higher odds of multiple substance use (AOR = 2.1; *p* = 0.02), while among girls, the difference was not statistically significant (AOR = 1.4; *p* = 0.16). When compared to opposite-gender attracted youth, both-gender attraction was associated with significantly higher odds of multiple substance use for both boys (AOR = 2.8) and girls (AOR = 2.2), while those not having been in love had significantly lower odds (boys’ AOR = 0.5, girls’ AOR = 0.4). Girls not responding to the love item also had significantly lower odds of multiple substance use (AOR = 0.6, *p* = 0.04), but not boys (AOR = 0.9, *p* = 0.65).

## 4. Discussion

This study aimed to explore the association between different romantic attraction patterns and substance use across national representative samples of adolescents from eight European countries and regions with various geographical location, history and level of tolerance towards sexual minorities. Our findings indicate higher risks for both same- and both-gender attracted youth to engage in substance use. This pattern was observed for single (cigarette smoking, alcohol consumption, drunkenness, and cannabis use) and multiple substance use (any two or more of cigarette smoking, alcohol consumption, or cannabis use), thus supporting findings from existing international literature. The fact that the pattern of the odds ratios remained very similar after controlling for gender, country/region and relative family affluence suggests that the elevated vulnerability of SMY to be engaged in substance use is a universal phenomenon, at least across the eight investigated European countries. These findings imply that as well as differentiating between heterosexual and homosexual or bisexual orientation, separating monosexual (heterosexual or gay/lesbian) and plurisexual (bisexual) identities may also reveal health disparities. Expanding the investigation to those who have not been in love revealed that, to a certain extent, reporting being in love (irrespective the gender of the love partner) was associated with higher odds of substance use than not having been in love. These results can be integrated with the minority stress and romantic stress theoretical models.

### 4.1. Gender and Attraction

Differences in alcohol, drunkenness, cannabis and multiple substance use were found across the groups of same- and both-gender attracted boys and girls. Regarding alcohol and drunkenness, same-gender attracted boys were at somewhat (but not to a significant extent) higher risk than both-gender attracted boys. Among girls, both-gender attraction was associated with higher risk for all five substance use indicators compared to same-gender attraction. This finding is in line with other studies showing that bisexual or both-gender attracted youths are among the highest risk of all SMY groups in relation to substance use [8,20,24], and even they are not homogenous in terms of risk [57].

### 4.2. Socioeconomic Status, Country and Attraction

Both family affluence and country/region were significant predictors in all multivariate models, however both unadjusted and adjusted substance use models follow very similar patterns. This indicates that the association between romantic attraction and substance use is not substantially influenced by family background or country/region of residence, at least in the eight European countries featured in our study. While lower socioeconomic status is associated with greater likelihood of reporting substance use [58], some argue that minority stress exacerbates the involvement of all SMYs in substance use, against which racial (or socio-economic) factors are not protective [11]. In other words, sexual minority status may be more strongly associated with substance use than socio-economic status. The observation that the risks of substance use among SMY is similar across regions and countries reinforces existing findings in this area, from single European countries [59,60,61,62,63] or from cross-cultural investigations [21].

### 4.3. Love: A ‘Sweet Poison’?

In line with our hypotheses, we have observed that those young people who reported never having been in love had significantly lower (0.4–0.7 times) odds of any single or multiple substance use than their opposite-gender attracted peers. This was found in the general models as well as in those disaggregated for gender. In other words, it seems that never having been in love is protective against substance use. This finding is in line with available evidence that romantic relationships (irrespective of the gender of the partner) may be associated with stress in adolescence [64], thus support the concept of romantic stress. Involvement in romantic relationships may be stressful for young people for a variety of reasons. These include separation and individuation from family, cultural expectations, conflicts with the romantic partner, and a double standard of love and sexual initiation for boys and girls [41,65]. As mentioned in Section 1.4, some participants might have been engaged in substance use (e.g., alcohol and drug consumption) to cope with romantic stress [41]. The romantic partner’s substance use habits may also be predictive of the adolescents’ own substance use [66]. Time spent with peers is predictive of substance use [67], and a similar association might be there with romantic partners. Further studies are needed to disentangle the separate effects of minority and romantic stress in SMY.

### 4.4. Non-Responders

Not responding to the item on love was not associated with higher odds of cigarette smoking, drunkenness, and cannabis use, but was associated with lower odds for alcohol use (in both boys and girls) and for multiple substance use (in girls) than for those who reported opposite-gender attraction. Adolescents may have various motives for not answering survey questions related to sexual orientation. They may be reluctant or unwilling to assume a socially stigmatized identity label, or it may reflect personal, cultural, religious, or political resistance to being categorized or defined by their sexuality [42]. We do not know participants’ motives for not responding to the item, but we speculate that this is not associated with elevated stress (based on the above-mentioned findings on the association of stress and frequent substance use). When developing the item on love, a few young people told our research team that they felt it is too private [68]. Further qualitative studies are needed to better understand young people’s motivations for not answering questions of this nature.

### 4.5. Limitations

Our findings are limited by the fact that the study was cross-sectional, therefore no causal or temporal inferences can be made. Given the low subsample sizes in countries, we could not carry out country-stratified analyses. However, from the fact that country as a control variable (as gender and family affluence) did not substantially change the pattern of the results, we infer that SMY may be at elevated risk regardless of their gender, family background and country of residence, at least in these eight European countries/regions.

We have concentrated on feelings of love for opposite-, same- or both-gender partners, which may not totally correspond to self-identified sexual orientation, erotic desire or sexual behavior [42,43].

Finally, we used a binary variable (boy or girl) to categorize adolescents’ gender, which does not reflect trans, non-binary or other gender minority groups. The links between gender, biological sex and sexual orientation constitute a very complex issue [69]. The HBSC International Network is currently working on how the survey can be more inclusive of both gender and sex diversity.

### 4.6. Reducing Risk and Promoting Resilience in Sexual Minority Youth

How can we reduce the risk associated with romantic attraction (compared to those who have not been in a romantic relationship) and the risk associated with both- and same-gender attraction compared to those who are attracted to opposite-gender partners? Love is experienced by many young people; professionals working with adolescents in health or social care, or educational settings, should be aware of the potential stressful effects of romantic relationships and minority stress, and be prepared to discuss these intimate matters with young people. Promoting healthy romantic relationships, both for SMY and heterosexual youth, may reduce (romantic) stress and have a positive impact on peer norms. Direct measures to promote health and resilience in SMY, such as ‘gay-straight alliances’ or ‘gender-sexuality alliances’ [70], media-based interventions to address sexual orientation related prejudice [71] or introducing safe school policies [72] have a documented beneficial effect on the health of not just SMY, but on heterosexual adolescents as well. Risk prevention and enhancing resilience and well-being in SMY should be part of national youth health strategies [73]. Some suggest that researchers and practitioners should consider how to shift from a victimizing and pathologizing narrative, which describes sexual minority individuals as ‘vulnerable’ [74]. A more positive view on sexual (and gender) minority people include, for instance, resilience, compassion, and tolerance towards members of other minorities [75]. Despite the hardships sexual minority young people experience, they have the potential to express their identity and love and lead healthy and happy lives.

### 4.7. Future Directions

Dimensions of sexual orientation, involvement in romantic attractions, minority stress, risk behaviors and psychosocial factors constitute a complex causal ‘web’. Future studies are needed to map how bullying involvement and social support shape substance use and other risky and health promoting behaviors, and various health outcomes in SMY. Using a positive approach, such research projects may also map health-protective factors and resources in sexual minority youth.

## 5. Conclusions

Sexual minority youth from eight European countries (identified based on reporting love for same- or both-gender partners) were found to be at higher risk of substance use behaviors than their opposite-gender attracted peers. On the other hand, adolescents who reported not having been in love were at lower risk of these substance use behaviors. These results support the assertion that romantic experiences on their own might be stressful for adolescents across different cultures, and that sexual minority status is associated with higher risk of substance use even after controlling for country/region, socio-economic status and gender. Targeted policy actions are needed to reduce risk and promote well-being and resilience in SMY, and further cross-national research needs to be conducted to better understand how dimensions of sexual orientation impact the health of young people.

## Figures and Tables

**Table 1 ijerph-16-03063-t001:** Characteristics of the sample, overall and by romantic attraction (*n* = 13,580).

	OVERALL		Opposite-Gender Love		Same-Gender Love		Both-Gender Love		Not in Love		Not Responding		Assoc. ^2^
	*n*	%	*n*	% (RA) ^1^	*n*	% (RA)	*n*	% (RA)	*n*	% (RA)	*n*	% (RA)	
Love													
Opposite-sex love	11,024	81.2											
Same-sex love	219	1.6											
Both-sex love	248	1.8											
Not in love	1756	12.9											
Not responding	333	2.5											
Country													
Belgium (French)	1779	13.1	1464	13.3	27	12.3	25	10.1	222	12.6	41	12.3	*p* < 0.001,
Bulgaria	1542	11.4	1306	11.8	61	27.9	38	15.3	115	6.5	22	6.6	*V* = 0.173
Switzerland	1692	12.5	1503	13.6	9	4.1	21	8.5	149	8.5	10	3.0	
England	1442	10.6	748	6.8	23	10.5	37	14.9	578	32.9	56	16.8	
France	1658	12.2	1365	12.4	34	15.5	33	13.3	207	11.8	19	5.7	
Hungary	1085	8.0	853	7.7	4	1.8	16	6.5	129	7.3	83	24.9	
Iceland	2980	21.9	2707	24.6	49	22.4	57	23.0	151	8.6	16	4.8	
North Macedonia	1402	10.3	1078	9.8	12	5.5	21	8.5	205	11.7	86	25.8	
Gender													
Boy	6732	49.6	5622	51.0	96	43.8	70	28.2	762	43.4	182	54.7	*p* < 0.001,
Girl	6848	50.4	5402	49.0	123	56.2	178	71.8	994	56.6	151	45.3	*V* = 0.080
Relative FAS													
Lowest 20 percent	2828	20.8	2217	20.1	67	30.6	63	25.4	393	22.4	88	26.4	*p* < 0.001,
Medium 60 percent	8127	59.8	6618	60.0	113	51.6	145	58.5	1070	60.9	181	54.4	*V* = 0.037
Highest 20 percent	2625	19.3	2189	19.9	39	17.8	40	16.1	293	16.7	64	19.2	

^1^ % (RA): Proportion within the given romantic attraction category. ^2^ Assoc.: Association between the given variable and romantic attraction.

**Table 2 ijerph-16-03063-t002:** Frequency of risk behaviors, overall and by romantic attraction.

	OVERALL		Opposite-Gender Love		Same-Gender Love		Both-Gender Love		Not in Love		Not Responding		Assoc. ^2^
	*n*	%	*n*	% (RA) ^1^	*n*	% (RA)	*n*	% (RA)	*n*	% (RA)	*n*	% (RA)	
Cigarette smoking in the last 30 days	13,504												
No	11,097	82.2	8933	81.5	150	69.4	164	66.4	1580	90.3	270	81.1	*p* < 0.001,
Yes	2407	17.8	2025	18.5	66	30.6	83	33.6	170	9.7	63	18.9	*V* = 0.105
Alcohol consumption in the last 30 days	13,440												
No	8348	62.1	6656	61.0	122	55.7	120	48.8	1214	70.0	236	70.7	*p* < 0.001,
Yes	5092	37.9	4251	39.0	97	44.3	126	51.2	520	30.0	98	29.3	*V* = 0.079
Drunkenness in the last 30 days	13,471												
No	11,680	86.7	9463	86.5	168	77.1	182	74.9	1587	90.8	280	87.0	*p* < 0.001,
Yes	1791	13.3	1477	13.5	50	22.9	61	25.1	161	9.2	42	13.0	*V* = 0.073
Cannabis use in the last 30 days	12,109												
No	11,159	92.2	9149	92.2	165	84.2	185	79.4	1522	94.5	138	91.4	*p* < 0.001,
Yes	950	7.8	770	7.8	31	15.8	48	20.6	88	5.5	13	8.6	*V* = 0.083
Multiple substance use in the last 30 days ^3^	13,580												
No	11,499	84.7	9262	84.0	165	75.3	173	69.8	1610	91.7	289	86.8	*p* < 0.001,
Yes	2081	15.3	1762	16.0	54	24.5	75	30.2	146	8.3	44	13.2	V = 0.097

^1^ % (RA): Proportion within the given romantic attraction category. ^2^ Assoc.: Association between the given variable and romantic attraction. ^3^ Any two of cigarette smoking, alcohol consumption or cannabis use in the last 30 days.

**Table 3 ijerph-16-03063-t003:** Crude and adjusted odds for the four types of substance use, overall and by gender.

	Univariate Model			Multivariate Model (Overall)			Multivariate Model Stratified for Gender					
	COR ^1^	*p*	(95% CI)	AOR ^2^	*p*	(95% CI)	AOR	*p*	(95% CI)	AOR	*p*	(95% CI)
Cigarettes in the last 30 days	(*n* = 13,504)			(*n* = 13,504)			Boys (*n* = 6693)			Girls (*n* = 6811)		
Opposite-gender love	1			1			1			1		
Same-gender love	**2.00**	<0.001	(1.48–2.69)	**1.85**	<0.001	(1.34–2.55)	**2.36**	<0.001	(1.46–3.82)	**1.57**	<0.001	(1.02–2.41)
Both-gender love	**2.28**	<0.001	(1.74–2.99)	**2.31**	<0.001	(1.71–3.13)	**2.80**	<0.001	(1.65–4.75)	**2.10**	<0.001	(1.44–3.04)
Not in love	**0.47**	<0.001	(0.40–0.56)	**0.44**	<0.001	(0.37–0.52)	**0.53**	<0.001	(0.41–0.70)	**0.38**	<0.001	(0.30–0.48)
Not responding	1.08	0.599	(0.81–1.43)	0.89	0.450	(0.67–1.20)	1.12	0.584	(0.75–1.65)	0.68	0.093	(0.44–1.07)
Alcohol in the last 30 days	(*n* = 13,440)			(*n* = 13,440)			Boys (*n* = 6693)			Girls (*n* = 6811)		
Opposite-gender love	1			1			1			1		
Same-gender love	1.27	0.093	(0.96–1.66)	1.20	0.230	(0.89–1.63)	**1.66**	0.036	(1.03–2.66)	0.97	0.880	(0.64–1.46)
Both-gender love	**1.67**	<0.001	(1.29–2.15)	**1.80**	<0.001	(1.33–2.43)	1.08	0.784	(0.64–1.81)	**2.15**	<0.001	(1.50–3.08)
Not in love	**0.67**	<0.001	(0.60–0.75)	**0.52**	<0.001	(0.46–0.59)	**0.56**	<0.001	(0.47–0.67)	**0.49**	<0.001	(0.42–0.58)
Not responding	**0.66**	0.001	(0.52–0.84)	**0.48**	<0.001	(0.38–0.62)	**0.57**	0.001	(0.41–0.79)	**0.37**	<0.001	(0.26–0.56)
Drunkenness in the last 30 days	(*n* = 13,471)			(*n* = 13,471)			Boys (*n* = 6693)			Girls (*n* = 6811)		
Opposite-gender love	1			1			1			1		
Same-gender love	**1.92**	<0.001	(1.39–2.67)	**1.81**	0.001	(1.28–2.55)	**2.42**	<0.001	(1.48–3.96)	1.38	0.204	(0.84–2.28)
Both-gender love	**2.20**	<0.001	(1.63–2.96)	**2.19**	<0.001	(1.59–3.02)	**1.93**	0.016	(1.13–3.31)	**2.25**	<0.001	(1.51–3.34)
Not in love	**0.65**	<0.001	(0.55–0.77)	**0.51**	<0.001	(0.43–0.61)	**0.55**	<0.001	(0.42–0.72)	**0.47**	<0.001	(0.37–0.60)
Not responding	0.98	.879	(0.70–1.36)	**0.68**	0.021	(0.48–0.94)	0.71	0.120	(0.46–1.09)	0.63	.090	(0.37–1.07)
Cannabis in the last 30 days	(*n* = 12,109)			(*n* = 12,109)			Boys (*n* = 5999)			Girls (*n* = 6110)		
Opposite-gender love	1			1			1			1		
Same-gender love	**2.21**	<0.001	(1.48–3.31)	**2.16**	0.001	(1.39–3.36)	**2.90**	0.001	(1.57–5.38)	1.62	0.142	(0.85–3.09)
Both-gender love	**3.19**	<0.001	(2.30–4.44)	**3.57**	<0.001	(2.48–5.13)	**4.12**	<0.001	(2.33–7.30)	**3.20**	<0.001	(1.99–5.15)
Not in love	**0.68**	0.001	(0.54–0.85)	**0.62**	<0.001	(0.48–0.79)	**0.71**	0.048	(0.51–1.00)	**0.53**	0.001	(0.36–0.77)
Not responding	1.18	0.581	(0.66–2.10)	1.05	0.979	(0.57–1.93)	1.10	0.821	(0.50–2.40)	1.01	0.990	(0.38–2.65)

^1^ COR: Crude odds ratios. ^2^ AOR: Odds ratios adjusted for region, gender, and relative family affluence. Boldface indicates statistically significant differences in (*p* < 0.05) odds for the given substance use in the given group, as compared to the reference group.

**Table 4 ijerph-16-03063-t004:** Crude and adjusted odds for multiple substance use, overall and by sex (*n* = 13,580).

	Univariate Model			Multivariate Model (Overall)			Multivariate Model Stratified for Sex					
							Boys (*n* = 6693)			Girls (*n* = 6811)		
	COR ^1^	*p*	(95% CI)	AOR ^2^	*p*	(95% CI)	AOR	*p*	(95% CI)	AOR	*p*	(95% CI)
Multiple substance use in the last 30 days ^3^												
Opposite-gender love	1			1			1			1		
Same-gender love	**1.79**	<0.001	(1.30–2.46)	**1.68**	0.003	(1.20–2.35)	**2.12**	0.002	(1.33–3.64)	1.39	0.161	(0.88–2.20)
Both-gender love	**2.34**	<0.001	(1.77–3.09)	**2.43**	<0.001	(1.79–3.31)	**2.78**	<0.001	(1.64–4.70)	**2.24**	<0.001	(1.54–3.27)
Not in love	**0.47**	<0.001	(0.39–0.56)	**0.44**	<0.001	(0.36–0.53)	**0.53**	<0.001	(0.40–0.70)	**0.38**	<0.001	(0.29–0.49)
Not responding	0.84	0.291	(0.60–1.16)	0.74	0.081	(0.53–1.03)	0.90	0.651	(0.58–1.41)	**0.59**	0.044	(0.35–0.99)

^1^ COR: Crude odds ratios. ^2^ AOR: Odds ratios adjusted for region, gender, and relative family affluence. Boldface indicates statistically significant differences in (*p* < 0.05) odds for the given substance use in the given group, as compared to the reference group. ^3^ Based on cigarettes, alcohol or cannabis in the last 30 days.

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
