# Peer review of "Romantic Attraction and Substance Use in 15-Year-Old Adolescents from Eight European Countries"

_ijerph, 2019, doi:10.3390/ijerph16173063_

Round 1

Reviewer 1 Report

Thank you for submitting this manuscript for consideration. I have really enjoyed reading your paper.

This paper set out to compare substance misuse in same and both-sex attracted 15-year-old adolescents from eight European Countries. This appears to have been achieved. 

The use of language in this paper is commendable. There is terminology being used around sexuality and sexual expression that is not used often or consistently within the existing literature but this is defined well. 

For me the thing that requires some clarity is how the eight countries were chosen to participate in the study. It is true that they are all European countries but the structure and agency is different in each, especially for SMY. I am not sure if this is fully addressed or explained. 

Generally, results are clear. Leaving choices to the discretion of teams leads to inconsistencies. Clarity is needed needed in lines 332-335. It would be stronger to ensure consistency of all participants and I am still unsure whether this has affected the results and my trust of them. 

Author Response

Response to Reviewer 1’s comments on IJERPH Manuscript No. ijerph-573585 titled “Romantic Attraction and Substance Use in 15-Year-Old Adolescents from Eight European Countries”

Thank you for submitting this manuscript for consideration. I have really enjoyed reading your paper.

This paper set out to compare substance misuse in same and both-sex attracted 15-year-old adolescents from eight European Countries. This appears to have been achieved.

The use of language in this paper is commendable. There is terminology being used around sexuality and sexual expression that is not used often or consistently within the existing literature but this is defined well.

For me the thing that requires some clarity is how the eight countries were chosen to participate in the study. It is true that they are all European countries but the structure and agency is different in each, especially for SMY. I am not sure if this is fully addressed or explained.

Dear Reviewer,

We would like to thank you for the valuable comments.

The HBSC survey in 2014 included 42 countries. The item on romantic attraction used in this present manuscript was optional, which meant it was additional to a core questionnaire that has been used by every country involved in HBSC. As with all other optional items, it was at the full discretion of the national research teams in each country to decide whether they include the item or not. Since HBSC strives to get a comprehensive picture of various dimensions of adolescent health (but we have only one classroom session to let children fill in the questionnaire), the number of the items that can be added to the questionnaire is very limited and this is the main reason why other countries could not administer the item in their national questionnaire. We have extended the Materials and Methods section to reflect the above:

Pages 5–6, Lines 245–247 and 253–255: “There were 42 countries that collected data as part of the HBSC international survey in 2014. Out of these, data from eight countries and regions are featured in this paper. (…) In the present study, substance use was monitored using mandatory items, while romantic attraction was measured by an optional item. As such, the measure of romantic attraction was included in the national surveys if the research team in the given country or region considered investigating the health of sexual minority youth substantially important.

In line with your comments we have extended the Introduction by outlining the large geographical, historical and tolerance-wise differences in these eight countries in Section 1.4. (Minority and romantic stress):

Pages 3-4; Lines 145–155: “The countries involved in our study represent large variation both geographically (from Iceland to North Macedonia), historically (from traditionally Capitalist countries as Belgium, England, France and Switzerland to post-Communist countries as Bulgaria and Hungary), and in terms of tolerance towards sexual minorities. The latter can be demonstrated by the International Lesbian Gay, Bisexual, Trans and Intersex Association’s Rainbow Score (https://www.ilga-europe.org/rainboweurope), a composite measure reflecting the legal situation and acceptance of gender and sexual minorities in different countries, ranging from 0 (gross violations of human rights) to 100 (full respect of human rights, full equality between sexual and gender minority and heterosexual and cisgender individuals). In the eight countries or regions involved in the present study, the Rainbow Score in 2014, when the data were collected, ranged from 13% in North Macedonia to 82% in the United Kingdom[39].”

Generally, results are clear. Leaving choices to the discretion of teams leads to inconsistencies. Clarity is needed needed in lines 332-335. It would be stronger to ensure consistency of all participants and I am still unsure whether this has affected the results and my trust of them.

Due to the process of the data collection procedures in HBSC (sampling is based on national educational statistics), it is not required to use weighting, if the actual sample to correspond with the sampling frame. In our case, only the French data needed correction.

Page 9; Lines 343–345: “Data were not weighted if the characteristics of the actual sample corresponded to those of the national sampling frame (e.g., gender or family affluence distribution). The only exception to this was France.”

Reviewer 2 Report

Thank you for the opportunity to review ‘Romantic attraction and substance use in 15-year-old adolescents from eight European countries’. This is a complex but well-conceived and well-implemented study which is really a kind of composite analysis of data available from eight different nations. I have to say that the more times I read this paper the better I liked it, which suggests that the authors may wish to spend a little more time in the Introduction telling the reader what to expect from the paper, briefly setting out arguments, aim and purpose. The disparate nature of environmental factors such as culture make it difficult to draw large conclusions from the study, and, as the authors point out (line 495) no causal or temporal inferences can be made from the study. Nevertheless, the study is useful and novel in its findings that SMY who have been ‘in love’ are at increased risk for polysubstance use. The authors appropriately note limitations for the study, although they don’t explicitly state the obvious, ipso hoc, non ergo propter hoc: just because two things appear to be related doesn’t mean that they are. There may be other unknown or non-specified influences that contributed to these associations, or that each may be related to a third unknown factor. For instance, the notion of transgressive behaviour is not considered anywhere in the literature—once you’ve transgressed the social or cultural prohibition against falling in love with a partner of the same sex, is it easier to transgress other social prohibitions such as substance use in young people? It is disappointing that these kinds of studies usually take a problematizing approach to both substance use and sexuality; what is astonishing is that despite social stigma, bullying, and legal and religious proscriptions that young people find their ways to express identity and love at all.   Nevertheless, the study is well-founded in theory and in data.

General issues

While I appreciate the authors’ diligence in including such an extensive, relevant and contemporary literature review, it takes up lines 59-212, about 28% of what is already quite a long and methodologically complex paper. I encourage the authors to summarise and shorten as much of the literature review, use only what is essential to create their argument/hypothesis, and to move more quickly to the heart of their paper. A way to do this would be to focus on literature related to key findings, and reserve less directly relevant literature for another paper. A way to frame the lit review would be to ask why the present study is necessary, important or innovative.

Similarly, while I understand the authors’ enthusiasm for their study—and this is a very complex study—it needs more to hold it together than “We aimed to describe and compare…” (line 214). That would makes this paper simply a ‘data dump’. As we see from the literature review, the hypothesis is in itself not especially novel, so what the reader seeks is what the essential, core, astonishing or interesting features of this paper are that will keep the reader’s attention? Other than the methodology, what is it about this paper that contributes to the development of new understandings, new theory, new practices? The authors make the argument that it is love itself which is the culprit (section 4.3). That is novel, and that is what the authors may wish to add into the Introduction, and perhaps even as a revised title of this paper. It would certainly make an interesting and engaging focus of the paper, rather than as a kind of aside, where it now appears.

Ethical considerations were well managed and well-addressed in this study.

Methodologically the study appears sound, although I acknowledge my lack of familiarity with ridit procedures, and the editors may wish to have this aspect of the analysis considered by someone with more relevant technical knowledge.

The findings are presented appropriately, the text refers to all the data tables, and each table is a useful contribution to the text.

The Discussion section is relevant and based on findings; the authors finally come to the ‘punch line’ in section 4.3. This is the engaging core of the Discussion section, and I hope, as I’ve noted above, the authors can foreshadow this in the Introduction. This is the novel contribution of this paper. (I might also add that this finding is consistent with my observation above about transgression. Just noted en passant.)  The section on practice interventions (paragraph 4.6) is, in my view, the weakest section of this paper, but interventions do not seem to be a focus for these authors. Nowhere in this paragraph is there talk of reduction of social stigma, legal recognition of same-sex partnerships by the state and by religious institutions, as well as school boards and committees, and such environmental changes would go far to normalising and integrating same-sex and ‘plurisexual’ partnerships and relationships. That said, I think the continued use of minority and romantic stress models in contemporary studies may be to do young people a significant injustice: at least in developed nations, many young people do not experience stress about minority sexual relationships until they encounter adult pressures. This is not to be naïve, or to ignore the realities of bullying (both online and in person), but in many (but certainly not all) places the world has moved forward on SMY and adults. That, however, would be a topic for quite a different paper.

Specific issues

Abstract, line 40, “who have not been attracted”—to whom or what?

Introduction, line 56, “First”. This is awkwardly placed, and does not scan well in English—it implies a list. I would omit the entire sentence.

Nomenclature: I fully realise the issue of taxonomy is contested, difficult, and inconsistent in the literature. Nevertheless, the authors must remain consistent within their own paper. The use of LGB youth (line 59 and throughout), SMY (line 53 and throughout), LGBT (line 78) is confusing. When is one used instead of another? If different acronyms are used (and I agree with the United Nations editorial use that people cannot be reduced to acronyms), then perhaps a brief explanation can be provided as to when one acronym is used instead of another? Or better, just remain consistent.

Iine 188, the syntax is garbled, and I don’t know what is meant.

Author Response

Response to Reviewer 2’s comments on IJERPH Manuscript No. ijerph-573585 titled “Romantic Attraction and Substance Use in 15-Year-Old Adolescents from Eight European Countries”

Thank you for the opportunity to review ‘Romantic attraction and substance use in 15-year-old adolescents from eight European countries’. This is a complex but well-conceived and well-implemented study which is really a kind of composite analysis of data available from eight different nations. I have to say that the more times I read this paper the better I liked it, which suggests that the authors may wish to spend a little more time in the Introduction telling the reader what to expect from the paper, briefly setting out arguments, aim and purpose. The disparate nature of environmental factors such as culture make it difficult to draw large conclusions from the study, and, as the authors point out (line 495) no causal or temporal inferences can be made from the study. Nevertheless, the study is useful and novel in its findings that SMY who have been ‘in love’ are at increased risk for polysubstance use. The authors appropriately note limitations for the study, although they don’t explicitly state the obvious, ipso hoc, non ergo propter hoc: just because two things appear to be related doesn’t mean that they are. There may be other unknown or non-specified influences that contributed to these associations, or that each may be related to a third unknown factor. For instance, the notion of transgressive behaviour is not considered anywhere in the literature—once you’ve transgressed the social or cultural prohibition against falling in love with a partner of the same sex, is it easier to transgress other social prohibitions such as substance use in young people? It is disappointing that these kinds of studies usually take a problematizing approach to both substance use and sexuality; what is astonishing is that despite social stigma, bullying, and legal and religious proscriptions that young people find their ways to express identity and love at all.   Nevertheless, the study is well-founded in theory and in data.

Dear Reviewer,

We would like to thank the valuable comments which we address one by one below. We understand your comment that “It is disappointing that these kinds of studies usually take a problematizing approach to both substance use and sexuality” and we fully agree with you in that other potential explanations, e.g., the ‘transgression’ concept you mentioned, should be part of the discourse on Sexual Minority Youth. Please note that the following text have been added to the Discussion (chapter 4.6, Reducing risk and promoting resilience in sexual minority youth):

Page 16; Lines 542–547: “Some suggest that researchers and practitioners should consider how to shift from a victimizing and pathologizing narrative, which describes sexual minority individuals as ‘vulnerable’[74]. A more positive view on sexual (and gender) minority people include, for instance, resilience, compassion, tolerance towards members of other minorities[75]. Despite the hardships sexual minority young people experience, they have the potential to express their identity and love and lead healthy and happy lives.”

To Section 4.7. on future research directions we have also added a sentence:

Page 16; Lines 551–552: “Using a positive approach, such research projects may also map health-protective factors and resources in sexual minority youth.”

General issues

While I appreciate the authors’ diligence in including such an extensive, relevant and contemporary literature review, it takes up lines 59-212, about 28% of what is already quite a long and methodologically complex paper. I encourage the authors to summarise and shorten as much of the literature review, use only what is essential to create their argument/hypothesis, and to move more quickly to the heart of their paper. A way to do this would be to focus on literature related to key findings, and reserve less directly relevant literature for another paper. A way to frame the lit review would be to ask why the present study is necessary, important or innovative.

We have substantially shortened the lit review and emphasized throughout that few and sporadic evidence exists outside North America. We believe our study is novel in the sense that it investigates young people from European countries and regions with different levels of acceptance towards sexual minorities. The following sentence has been added to the introduction:

Page 2; Lines 52–61: “The studies show a large variation on using (biological) sex or (socially constructed) gender. They also employ various sexual identity terms or classify youth based on other dimensions of sexual orientation, such as gender of sexual or love partner(s). In this study, we use the term ‘gender’, as we assigned respondents based on whether they are boys or girls. The ‘sexual minority youth’ (SMY) term is used, as this is the most inclusive, unless we refer to studies that used more specific terminology (such as LGB).

Extensive research indicates that SMY are more likely to engage in substance use[3,4]. However, the validity of the evidence is limited by the fact that most investigations have been conducted in North America. There are just a few sporadic observations from other countries, and cross-cultural comparisons are largely missing.”

Similarly, while I understand the authors’ enthusiasm for their study—and this is a very complex study—it needs more to hold it together than “We aimed to describe and compare…” (line 214). That would makes this paper simply a ‘data dump’. As we see from the literature review, the hypothesis is in itself not especially novel, so what the reader seeks is what the essential, core, astonishing or interesting features of this paper are that will keep the reader’s attention? Other than the methodology, what is it about this paper that contributes to the development of new understandings, new theory, new practices? The authors make the argument that it is love itself which is the culprit (section 4.3). That is novel, and that is what the authors may wish to add into the Introduction, and perhaps even as a revised title of this paper. It would certainly make an interesting and engaging focus of the paper, rather than as a kind of aside, where it now appears.

Thank you for highlighting that our notion – that love may be an explanation on its own – is novel. We have renamed Section 1.4. to reflect both minority and romantic stress and added:

Page 4; Lines 164–171: “Another potential explanation, partly overlapping with the minority stress model, is that love, irrespective of the gender of the partner(s) with whom a young person is in love with, may be associated with stress on its own. Indeed, a cross-cultural study conducted in 17 countries found that adolescents experienced stress related the romantic relationships, especially in Mid- and South-European countries. Overall, around 20% of the adolescents used externalizing coping strategies, such as alcohol and drug use, to cope with these stressors[41]. This prompts the notion that maybe not just same-or both-gender attracted adolescents may be at elevated risk of substance use, but anyone who are in love may be at higher risk than those who are not being in love.”

We have also updated other parts of the manuscript to reflect on this (e.g. Section 4.3).

We have also extended Section 1.4. (Aims of the present study), to highlight the diversity of the countries and regions featured in the analysis:

Pages 3–4; Lines 145–155: “The countries involved in our study represent large variation both geographically (from Iceland to North Macedonia), historically (from traditionally Capitalist countries such as Belgium, England, France and Switzerland to post-Communist countries as Bulgaria and Hungary), and in terms of tolerance towards sexual minorities. The latter can be demonstrated by the International Lesbian Gay, Bisexual, Trans and Intersex Association’s Rainbow Score (https://www.ilga-europe.org/rainboweurope), a composite measure reflecting the legal situation and acceptance of gender and sexual minorities in different countries, ranging from 0 (gross violations of human rights) to 100 (full respect of human rights, full equality between sexual and gender minority and heterosexual and cisgender individuals). In the eight countries or regions involved in the present study, the Rainbow Score in 2014, when the data were collected, ranged from 13% in North Macedonia to 82% in the United Kingdom[39].”

Ethical considerations were well managed and well-addressed in this study.

Please note that we have added one more sentence to emphasize that our study was conducted in line with the WHO standards:

Page 5, Lines 285–287: “Our research procedures are following the WHO Standards and operational guidance for ethics review of health-related research with human participants (https://www.who.int/ethics/research/en/).”

Methodologically the study appears sound, although I acknowledge my lack of familiarity with ridit procedures, and the editors may wish to have this aspect of the analysis considered by someone with more relevant technical knowledge.

The findings are presented appropriately, the text refers to all the data tables, and each table is a useful contribution to the text.

The Discussion section is relevant and based on findings; the authors finally come to the ‘punch line’ in section 4.3. This is the engaging core of the Discussion section, and I hope, as I’ve noted above, the authors can foreshadow this in the Introduction. This is the novel contribution of this paper. (I might also add that this finding is consistent with my observation above about transgression. Just noted en passant.)  The section on practice interventions (paragraph 4.6) is, in my view, the weakest section of this paper, but interventions do not seem to be a focus for these authors. Nowhere in this paragraph is there talk of reduction of social stigma, legal recognition of same-sex partnerships by the state and by religious institutions, as well as school boards and committees, and such environmental changes would go far to normalising and integrating same-sex and ‘plurisexual’ partnerships and relationships. That said, I think the continued use of minority and romantic stress models in contemporary studies may be to do young people a significant injustice: at least in developed nations, many young people do not experience stress about minority sexual relationships until they encounter adult pressures. This is not to be naïve, or to ignore the realities of bullying (both online and in person), but in many (but certainly not all) places the world has moved forward on SMY and adults. That, however, would be a topic for quite a different paper.

Again we thank you for these insights. While many of these cannot unfortunately be addressed as they are out of the scope of our paper, we have updated the discussion, Sections 4.6 and 4.7 to reflect the need of a more positive approach:

Page 16; Lines 542–547: “Some suggest that researchers and practitioners should consider how to shift from a victimizing and pathologizing narrative, which describes sexual minority individuals as ‘vulnerable’[74]. A more positive view on sexual (and gender) minority people include, for instance, resilience, compassion, tolerance towards members of other minorities[75]. Despite the hardships sexual minority young people experience, they have the potential to express their identity and love and lead healthy and happy lives.”

Page 16, Lines 552–553: “Using a positive approach, such research projects may also map health-protective factors and resources in sexual minority youth.”

Specific issues

Abstract, line 40, “who have not been attracted”—to whom or what?

Changed to “who have not been in love” in line with your other comments.

Introduction, line 56, “First”. This is awkwardly placed, and does not scan well in English—it implies a list. I would omit the entire sentence.

We have deleted the entire sentence.

Nomenclature: I fully realise the issue of taxonomy is contested, difficult, and inconsistent in the literature. Nevertheless, the authors must remain consistent within their own paper. The use of LGB youth (line 59 and throughout), SMY (line 53 and throughout), LGBT (line 78) is confusing. When is one used instead of another? If different acronyms are used (and I agree with the United Nations editorial use that people cannot be reduced to acronyms), then perhaps a brief explanation can be provided as to when one acronym is used instead of another? Or better, just remain consistent.

We have added to the introduction:

Page 2, Lines 52–57: “The studies show a large variation in the use of (biological) sex or (socially constructed) gender. They also employ various sexual identity terms or classify youth based on other dimensions of sexual orientation, such as gender of sexual or love partner(s). In this study, we use the term ‘gender’, as we assigned respondents based on whether they are boys or girls. The ‘sexual minority youth’ (SMY) term is used, as this is the most inclusive, unless we refer to studies that used more specific terminology (such as LGB).”

Iine 188, the syntax is garbled, and I don’t know what is meant.

Reformulated to (now Page 4, Lines 189–190): “Health disparities in SMY can be found when respondents are classified by same- or both-gender romantic attraction.”